# Individual, Family, and Environmental Correlates of Motor Competence in Young Children: Regression Model Analysis of Data Obtained from Two Motor Tests

**DOI:** 10.3390/ijerph17072548

**Published:** 2020-04-08

**Authors:** Donna Niemistö, Taija Finni, Marja Cantell, Elisa Korhonen, Arja Sääkslahti

**Affiliations:** 1Faculty of Sport and Health Sciences, University of Jyväskylä, 40014 Jyväskylä, Finland; taija.finni@jyu.fi (T.F.); leaelisa.korhonen@gmail.com (E.K.); arja.saakslahti@jyu.fi (A.S.); 2Department of Inclusive and Special Needs Education, University of Groningen, 9712 Groningen, The Netherlands; m.h.cantell@rug.nl

**Keywords:** socioecological model, locomotor skills, ball skills, balance skills, coordination, TGMD-3, KTK, temperament

## Abstract

Physical activity and motor competence (MC) have been considered to be closely related and prevent childhood obesity. The aim of the study was two-fold: to examine MC measured with two different tools in relation to individual, family, and environmental correlates and to investigate gender differences in MC. The Test of Gross Motor Development-Third Edition (TGMD-3) was administered to three- to seven-year-old children (*n* = 945), while the Körperkoordinationstest für Kinder (KTK) was also used for five- to seven-year-old children (*n* = 444). The parent questionnaire (*n* = 936) included questions about individual (e.g., participation in organized sports), family (e.g., parents’ education level), and environmental (e.g., access to sports facilities) correlates. The children’s temperament was assessed using the Colorado Childhood Temperament Inventory (CCTI) questionnaire. Data were analyzed using one-way analysis of variance and linear mixed-effects regression models. The regression models explained 57% and 38% of the variance in TGMD-3 and KTK, respectively. Individual correlates, including older age, more frequent participation in sports, and specific temperament traits of activity and attention span-persistence, were the strongest predictors for better MC. Small gender differences were found in both assessment tools, albeit in a different manner. In conclusion, socioecological correlates of MC in young children are multidimensional, and individual correlates appear to be the most important predictors of MC. Importantly, the correlates can differ according to the MC assessment tools.

## 1. Introduction

Motor competence (MC) and physical activity (PA) have been found to be closely and bidirectionally related in several theoretical models [1,2,3] and studies [4]. Consequently, it is claimed that if a child is not physically active, he/she may have a greater tendency to have lower MC, or vice versa, and be at risk of gaining unhealthy body weight [5]. Children’s weight status negatively influences their future level of gross motor coordination, and vice versa [6]. According to a recent systematic review [7], PA begins to decline from early childhood. This review pointed out that during the life course, inactivity is more persistent; therefore, intervening during early years is extremely important. In fact, the age under seven years can be defined as a rapid period for the development of fundamental movement skills, known as MC [8,9].

During infancy, a child’s development is evaluated almost exclusively by motor development [10]. New motor behaviors emerge from a mix of interacting factors [11]. Some of these motor behaviors are less recognized to be directly linked to motor behavior (e.g., facial expressions and speech), while others are known to be important milestones for a child’s overall development (e.g., walking) [11]. MC develops together with biological maturation [12]. However, the thought that as most children are naturally curious and love to play and explore, these skills are learned easily [13], and without practice is often misleading.

Motor development involves the acquisition and refinement of basic patterns via repetition [14], and these basic movement patterns form the foundation of more specialized and complex skills that a child will achieve later in life [3,8,14]. The mastery of MC is a prerequisite for daily life functioning and participation in physical or sports-specific activities later in life [13,15]. Moreover, it contributes to a balanced caloric intake, and contrarily, overweight children often have lower MC [16,17]. Therefore, PA plays a major role in providing these opportunities for repetition in children. 

The basic patterns of MC are divided into locomotor (LM) skills (e.g., running, jumping, and galloping), stability movement skills (e.g., turning, balancing, and bending), and finally, manipulative movement skills (e.g., throwing, catching, and kicking) [8], henceforth called ball skills (BS). Each of these skills plays a specific role in a child’s development. In particular, LM skills are important for enabling (independent) movement, leading to increased opportunities to engage in social and cognitive interactions [18] in the environment [11]. BS are crucial for hand–foot coordination [11] and its development, and stability helps children maintain balance on variable and unsteady surfaces. All these skills are associated with MC development, which has other great benefits for a child’s general health and wellbeing. MC is positively associated with cognitive functions, such as executive functions [19], attention and working memory, information processing speed [20,21,22,23], as well as language development [24], reading [25], and psychological functions [10]. These associations have been explained through similar maturation schedules of the brain structures controlling motor and cognitive functions for which children’s active interaction with their environment is crucial [26]. 

As motor development is a multidimensional process [11,27], it differs according to the motor skill category, e.g., [27,28]. In this study, two MC assessment tools were used to provide broader knowledge about the range of correlates of MC in young children. According to the socioecological model [29], a child’s behavior stems from reciprocal interactions among individual, family, environmental, and community levels. According to Barnett et al. [27], correlates that are directly associated with the individual level seem to be the most important ones for MC. However, other factors related to a child’s life and surroundings may enhance or limit the possibilities of PA and MC development [29,30]. For example, a child’s PA or MC has been found to be positively associated with his/her father’s [31] or mother’s PA behavior [32]. Additionally, Laukkanen et al. [33] recently found that children with an agreeable temperament (individual correlate, biological characteristic) had more parental support for PA (family correlate). In contrast, children with a less agreeable temperament had less opportunities to participate in PA with their parents and received less frequent parental support for PA. Such a lack of family support may be related to the interaction style stemming from the child’s more or less demanding temperament. It is not yet well researched, but such individual correlates can be associated with lower PA levels and therefore with lower MC levels.

The choice of two assessment tools makes it possible to examine the gender differences found in some studies [12,34,35] based on divergent skill categories, wherein most studies found that boys had a better gross motor index than girls [12,35]. However, it is suggested that the gender differences in early childhood are not based on biological factors [15], but are more likely related to family, environmental, and sociocultural contexts [35,36,37,38]. Therefore, we consider it important to assess MC using two assessment tools covering divergent aspects of MC and to include correlates belonging to all three levels (individual, family, and environmental) in the analysis. Sallis et al. [30] stated that to be able to make substantial behavioral changes, interventions must target changes at each level of the socioecological model.

Although PA, MC, and body mass index (BMI) are closely related [39], the younger the child, the more dependent his/her (motor) development and daily activities are on his/her family environment. In this equation, individual correlates along with family and environmental correlates play a role in the actual and long-term PA levels of children, influencing children’s motor development. As the majority of the studies [12,34,35,36] focus on individual correlates of MC, we wanted to broaden the research into a socioecological perspective, such as family and environmental correlates, to gain more knowledge about MC development. Thus, the aims of this study were to examine correlates associated with MC in three- to seven-year-old Finnish children using two internationally well-known MC assessment tools and to examine whether there were gender differences in MC.

## 2. Materials and Methods 

The Ethics Committee of the University of Jyväskylä, Finland, granted approval for the study on 31 October 2015 (Skilled Kids, 31.10.2015). The parents of the participating children provided their written consent. The children were informed about all the study procedures and their right to opt out of the study at any time, without consequences.

### 2.1. Study Protocol and Participants

The aim of the Skilled Kids study was to have a nationally representative sample of 1000 children aged 3 to 7 years from Finnish childcare centers. The sample was recruited on the basis of the Finnish National Registry of Early Educators, which included 2600 childcare centers. Based on this registry, cluster random sampling was performed, i.e., childcare centers were randomly chosen from the metropolitan area and southern, central, and northern Finland based on postal codes. The number of childcare centers involved in one region was weighted with the population density of the area. The recruitment took place in the autumn of 2015. Thirty-seven childcare centers participated in total: six from the metropolitan area, eleven from southern Finland, thirteen from central Finland, and seven from northern Finland. Ten childcare centers (27%) declined to participate due to a lack of space, interest, time, or a low number of children. The aim of the Skilled Kids study and the recruitment process have been described in detail in previous studies [33,40,41]. For the recruited childcare centers, the respective directors first approved the participation, and the staff was informed about the study. In total, 1239 children (78.5%) received consent for study participation. The measurements were conducted in childcare center settings between November 2015 and September 2016 by two researchers (D.N. and A.S.), along with two research assistants.

The study sample consisted of 945 children (mean age 5.42 years, boys = 473 (50.1%)); however, the number of participants differed for different MC assessment tools. The detailed descriptive data of the study sample are provided in Table 1.

### 2.2. Motor Competence

MC was measured using two different assessment tools. Children aged 3 to 7 years were assessed using the Test of Gross Motor Development-Third Edition (TGMD-3) [42,43]. Moreover, children aged 5 to 7 years (n = 444, mean age 6.2 years, boys = 234 (52.7%)) completed an additional MC test, the Körperkoordinationstest für Kinder (KTK) [44].

First, all children aged 3 to 7 years completed the TGMD-3 measurements [42,43]. This process-oriented measurement evaluates the quality of the skills and has two skill categories concentrating on locomotion (LM) and on ball skills (BS). LM skills include a summary of six skills evaluated by the following points: run (0–8 points), gallop (0–8 points), hop (0–8 points), skip (0–6 points), horizontal jump (0–8 points), and slide (0–8 points), resulting in a maximum of 46 points. BS include a summary of seven skills: two-hand strike of a stationary ball (0–10 points), one-hand forehand strike (0–8 points), one-hand stationary dribble (0–6 points), two-hand catch (0–6 points), kicking a stationary ball (0–8 points), overhand throw (0–8 points), and underhand throw (0–8 points), resulting in a maximum of 54 points. An educated observer, who analyzed the skills according to the fulfilment of the given criteria (three to five criteria for one skill), evaluated each skill (0 points if the given criteria were not fulfilled and 1 point if they were fulfilled). Children performed each skill twice, and the recorded score was the sum of the received points of these two performances (maximum of 2). The TGMD-3 gross motor index was the sum of LM skills and BS, with a theoretical maximum of 100 points. The gross motor index itself is the most reliable test score [42].

The TGMD-3 protocol was carefully followed according to the manual described previously [33,40]. TGMD-3 has been demonstrated to have a good to excellent intra-rater and inter-rater reliability [43], and it is valid and reliable both internationally [13,45] and in the Finnish context [33,40]. Within this study sample, the inter-rater reliability of the TGMD-3 gross motor index was 0.88 (95% confidence interval (CI) = 0.85–0.92), tested among 167 children [40].

To have complementary information about gross motor coordination and body control of children aged 5 to 7 years, they participated also in the KTK assessment. In this product-oriented assessment tool, evaluation is based on the total score of the four items included in the test battery, and as the test is result-based, the theoretical total maximum points cannot be specified. The test items include balance, with a walk of eight steps backwards on balance beams (width 6.0 cm, 4.5 cm, and 3.0 cm; maximum score of 72 points), hopping on one leg over an obstacle (maximum score of 78 points), jumping laterally from side to side on a jumping base for 15 s (the sum of the number of correct jumps in two trials), and shifting platforms as quickly as possible for 20 s (the sum of the number of points in 20 s in two trials). Each skill was performed and observed carefully following the manual instructions by experienced observers. Finally, the sum of these latter scores yielded the total sum score for the KTK test. The raw score was used in the present analysis, as recommended [46,47]. The KTK assessment tool is considered to be highly reliable internationally, most likely because it is result-based, with the test-retest reliability coefficient of the total score being 0.97 and the subtests ranging between 0.80 and 0.96 [44].

### 2.3. Individual Correlates

#### 2.3.1. Biological Correlates 

Each child’s exact age was calculated on the basis of the date of birth related to the date of assessments, with 1 month accuracy. In the tables, however, the age is reported in years following common convention. The children’s weight (Seca 877) and height (Charder HM 200P) were directly measured, with the accuracy of a decimal. BMI was calculated as weight/height^2^ (kg/m^2^) and converted to BMI standard deviation scores (BMI SDS) using the most recent national BMI references [48].

The child’s temperament was assessed using a parental rating instrument, the Colorado Childhood Temperament Inventory (CCTI). This questionnaire is suitable for children aged up to 7 years [49]. It involves six dimensions of personality: sociability, emotionality, activity, attention span-persistence, reaction to food, and soothability. Each scale was constructed using five more specific statements. One statement representing each scale is presented here: sociability, “Child makes friends easily”; emotionality, “Child gets up easily”; activity, “Child is very energetic”; attention span-persistence, “Child plays with a single toy for long periods of time”; reaction to food, “Child rarely took a new food without fussing”, and soothability, “Whenever the child starts crying, (s)he can be easily distracted”. As a response, each parent had to rate every statement from 1 (“not at all like the child”) to 5 (“a lot like the child”). In total, there were 30 statements, five in every scale; therefore, the maximum points for each scale were 25 (5 × 5). The validity of CCTI is reported to be good, and its reliability is moderately high [49]. Scale scores were used in the analyses.

#### 2.3.2. Behavioral Correlates 

The sedentary time, time spent outdoors, and participation in organized sports were assessed through a parental questionnaire. The test-retest reliability of all the items was investigated with 30 responses (obtained over 21 days); these are marked in parentheses with an intra-class correlation coefficient (ICC) and a 95% confidence interval (CI) after each item. The sedentary time (ICC = 0.45; 95% CI −0.09–0.80) was assessed through the following questions: “Think about your child’s typical day and situations when (s)he is sitting or lying down or is sedentary in some other way (e.g., in a car, in a sandbox or in a trolley, in front of the TV or while playing with a puzzle). For how long, at the most, does such a sedentary activity approximately last continuously and without breaks?” (1 = >15 min, 2 = 30 min, 3 = 60 min and 4 = ≥90 min) and “How often is your child engaged in long and continuous sedentary activities in a day?” (1 = once, 2 = twice or thrice, 3 = four to five times and 4 = ≥six times). The amount of sedentary time (min) in a day was calculated using the aforementioned information (min/time × times/day). The time spent outdoors (ICC = 0.62; 95% CI = −0.12–1.0) was obtained by asking “How much time, on average, does your child spend outdoors after a preschool day/on weekends?” The scale for weekdays ranged from 0 to 3 (0 = not at all, 1 = under 30 min/day, 2 = approximately 30–60 min/day, and 3 = over 60 min/day), while the scale for weekends ranged from 0 to 4 (0 = not at all, 1 = under 30 min/day, 2 = approximately 30–60 min/day, 3 = 1–2 h/day, and 4 = over 2 h/day). The total score from both scales was used to represent the time spent outdoors, with 7 being the maximum score. Participation in organized sports (ICC = 0.81; 95% CI = 0.60–0.91) was assessed by asking the following: “Does your child participate in organized PA or sports in a group or a sports club?” If the answer was “yes”, further questions regarding such activities were asked, as follows: “How many times a week?” and “For how many minutes at a time?” The total time (min) spent on organized sports per week was calculated and used in the analyses. 

### 2.4. Family Correlates

Due to divergent family backgrounds, we used the concepts of respondent and partner instead of mother or father. Later, female respondents were called mothers and male respondents were called fathers. The parents’ mean education level was the mean value of the respondent’s (ICC = 0.96; 95% CI = 0.92–0.98) and partner’s (ICC = 0.94; 95% CI = 0.87–0.97) education level (1 = comprehensive school, 2 = high school/vocational school, 3 = polytechnic, and 4 = university). Each respondent’s PA (ICC = 0.72; 95% CI = 0.50–0.86) was divided on a scale from 0 to 4 (0 = not at all, 1 = randomly few times a month, 2 = approximately once a week, 3 = twice or thrice a week, and 4 = over four times a week).

### 2.5. Environmental Correlates

The number of electronic devices (varying from ICC = 0.80; 95% CI = 0.62–0.90 to ICC = 1.00) was assessed through the following question: “Does your child have access to any or some of the following: (a) TV, (b) game console, (c) computer, (d) smartphone, tablet, iPad or any other smart device or (e) something else (if yes, then what)?” Finally, the child’s access to sports facilities (varying from ICC = 0.52; 95% CI = 0.21–0.74 to ICC = 0.87; 95% CI = 0.75–0.94) was asked, e.g., “Evaluate how often your child has used sport or outdoor facilities situated in your own locality or nearby municipality”. The questionnaire included 10 divergent and organized sports facilities (e.g., playing field, playground, swimming hall, and indoor sports hall) and an open space for the facilities that were being used, but were not listed. The use of each facility was scored on a scale from 0 to 4 (0 = no access to a facility, 1 = nearly never, 2 = randomly, 3 = weekly, and 4 = approximately daily). Additionally, the respondents were asked the following: “Is there a large area for the child’s free play in your home yard (front- or backyard, garden, etc.)?” (no = 0 point and yes = 1 point) and “How often is your child allowed to play in the yard?” The frequency was scored from never to nearly daily (0 = never, 1 = during weekends, 2 = every now, and then and 3 = nearly daily). The total access to sports facilities and to the home yard was calculated by adding all the respondents’ scores, with the maximum points being 44.

### 2.6. Statistical Analysis

IBM SPSS Version 24.0 (IBM Corp., Armonk, NY, USA) was used for the analyses, and the level of significance was set at *p* < 0.05. Descriptive statistics (mean and standard deviation (SD)) were calculated for all variables (Table 1). Gender differences were calculated using a *t*-test (Table 1). A linear regression model with the enter method was used to analyze the associations between the individual, family, and environmental correlates and the two MC assessment tools (TGMD-3 and KTK). However, as childcare centers may be associated with MC [50], the goodness of fit was tested with and without a childcare cluster. In all the models, the goodness of fit was significantly better when linear mixed-effects models with a childcare cluster were used (for all models, *p* < 0.001). Therefore, in the final analyses, linear mixed-effects models were used. In Model 1 for TGMD-3 and KTK, all the individual, family, and environmental correlates predicting TGMD-3 (Table 2) or KTK (Table 3) were entered. The least significant correlates were removed from Model 1 one at a time. Then, Model 1 was re-run with all the remaining correlates until there were only significant correlates left in the final model, Model 2. The order of removal from Model 1 is represented in Table 2 (TGMD-3) and Table 3 (KTK). This so-called backward method made it possible to consider the interdependency (mutual covariance) of predictors at each step of modelling. In Models 1 and 2, the number of items varied because of missing data in the remaining variables.

## 3. Results

### 3.1. Descriptive Results

All children were aged three to seven years (mean age 5.42 years, SD 1.12). The study sample had an equal distribution of girls (n = 472, 49.9%, mean age 5.4 years) and boys (n = 473, 50.1%, mean age 5.5 years). Of the parents (n = 936; mean age 35.8 years, SD = 5.4) who answered the parental questionnaire, most were mothers (n = 816, 87.2%). Two-thirds of the parents who responded to the questionnaire had polytechnic- or university-level education (n = 569, 60.7%). More detailed information about the study sample is provided in Table 1.

Regarding the gender differences, boys had lower LM skills, better BS, and a better TGMD-3 gross motor index than girls (Table 1). Boys also spent more time outdoors, used more electronic devices, and more frequently accessed sports facilities according to their parents. Some differences in temperament were noted between boys and girls. The parents described boys as being more active and girls as having higher attention span-persistence. The level of MC in the children increased with age in both assessment tools (Figure 1). 

### 3.2. Correlates of Test of Gross Motor Development-Third Edition 

In Model 2, having TGMD-3 as a dependent variable, children’s age (older), temperament traits, such as activity (higher) and attention span-persistence (higher), participation in organized sports (higher), and access to sports facilities (higher) explained 57% of the variance in the TGMD-3 gross motor index among children (Table 2). 

### 3.3. Correlates of Körperkoordinationstest für Kinder 

In Model 2 having KTK as a dependent variable, children’s age (older), gender (female), temperament traits such as emotionality (lower), activity (higher), and attention span-persistence (higher), participation in organized sports (higher), and parents’ mean education level (higher) explained 38% of the variance in the KTK total score among the children (Table 3). 

Of all the variables, a child’s age was most strongly associated with MC in both models; therefore, the older the child, the better MC they had. Furthermore, in both assessment tools, the participation in organized sports had a positive association with MC, in addition to having a more active temperament and having higher attention span-persistence as a dimension of personality.

## 4. Discussion

Based on the socioecological model, we investigated the associations of individual, family, and environmental correlates with MC in young children using the TGMD-3 gross motor index and KTK total score. We also examined whether there were any gender differences in the dependent variables.

Our study had several important findings. First, some individual (age, participation in organized sports, and temperament traits such as activity, attention span-persistence, and emotionality), family (parents’ education level), and environmental correlates (access to sports facilities) were associated with the children’s MC, supporting the socioecological model. Second, a model including six individual, family, and environmental correlates explained 57% of the variability in the TGMD-3 gross motor index, while seven correlates explained 38% of the variability in the KTK total score. Third, some gender differences were found; however, the findings of TGMD-3 and KTK were different.

In our study, higher variance in MC was explained with TGMD-3 than with KTK. This result may reflect the fact that these assessment tools measure different aspects of MC [13,51]. On the one hand, research suggests that TGMD-3 is a sports-centric test that uses quality evaluations of LM skills and BS, but lacks balance skills [13]. On the other hand, research suggests that KTK is a non-sports specific assessment tool [13,46] focusing on result-based evaluation of balance and motor skills, but lacks BS. Hence, although these two assessment tools complement each other, an assessment tool should ideally include a whole range of fundamental movement skills, i.e., balance skills, LM skills, and BS.

Other differences emerged in the assessment tools as some correlates, such as gender, emotionality of the child, parents’ education level, and access to sports facilities, were related in either, but not both assessment tools. These differences in the associated correlates may stem from the fact that the choice of the assessment tool provided varying outcomes as the tools measure different aspects of MC. Interestingly, the correlates that were significant in both TGMD-3 and KTK models (age, participation in organized sports, and temperament traits such as activity and attention span-persistence) were individual correlates, underlining the finding of Barnett et al. [27] that individual correlates seem to be the most important ones for MC development.

### 4.1. Individual Correlates of Motor Competence 

We found that older children tended to have better MC levels in both MC assessment tools. In line with previous studies, it was evident that age was a strong predictor of MC levels in children, affirming the role of age in MC [1,2,12,15,36,52]. This increase in MC in children aged three to seven years can be explained by the rapid biological development during these early years [53], wherein the high plasticity of the nervous system contributes to the major improvement in coordination [11,14]. However, children do not develop MC solely through maturational processes as coordinative movements need to be learned, practiced, and reinforced [54]. Therefore, regardless of age or gender, children should be encouraged to move and to develop age-appropriate MC [11,55].

This study provided novel information about the importance of temperament traits for motor development. More specifically, traits such as activity and attention span-persistence were found to be positively associated with MC using both motor assessment tools. This was a rather novel result, as the association between MC and temperament during early childhood is not yet widely understood. Temperament is rather stable [49,56] over time; thus, children who tend to have an active type of temperament, as well as children who show persistency when faced with challenges can be motivated and persistent in learning and rehearsing motor tasks. Interestingly, a recent study by Laukkanen et al. [33] demonstrated that children with an agreeable temperament (referring to a factor created from the total scores for sociability, activity, and attention span-persistence) tended to have more parental support for PA. Accordingly, there is evidence that the lack of fit between a child’s temperamental characteristics and parents’ responses [56] can influence the overall development of the child. Since the parent–child relationship is bidirectional, also the child’s behavior influences parenting [57]. In essence, it can be questioned if some children benefit from temperament traits such as activity and attention span-persistence not only in terms of motor development, but also in terms of the amount of parental support received for PA.

Additionally, children who were more emotionally regulated had a better KTK total score. This may also mean that during early childhood, as is consistently found in exercise psychology (e.g., emotion), regulating one’s effort control and distractibility during motor performance [56] may help maintain focus. Thus, temperament can be associated with motor development both directly and indirectly. The future research would benefit from a multidisciplinary collaboration between different professionals to better understand the role of a child’s temperament and parents’ behavior in the development of MC.

Participation in organized sports was associated with better scores in both assessment tools, underlining the fact that during early childhood, motor development benefits from sports-related hobbies [38,41,58] related to LM skills [27,41], BS [41], and coordination [55]. Although participation in organized sports is mainly positively related to MC development, there may be differences between different environments [41] and countries [59]. Therefore, one should not forget the importance of outdoor play and everyday life choices [59] that help to accumulate more daily PA [60].

### 4.2. Family and Environmental Correlates of Motor Competence 

A child’s access to sports facilities and parents’ educational level were associated with better TGMD-3 and KTK total scores. Access to sports facilities had a greater association; however, these associations were smaller than the aforementioned findings with other individual correlates. Some other studies have likewise demonstrated that family- and environmental-level correlates are not as closely associated with MC as individual correlates [27] or their significance to motor development is not easily demonstrated using the current assessment tools. However, there is evidence that having skill-related equipment or sports facilities near one’s home is positively associated with MC [31]. Thus, having a supportive environment in terms of toys and equipment may help develop MC or children with better MC may be provided more equipment [27] and opportunities to access sports facilities. Although the amount of toys and equipment was not assessed in the present study, we support the idea of Cools et al. [31] that providing access to sports facilities creates an environment conducive to MC development. Lastly, the higher the parents’ education level, the better was the KTK total score. This is in line with previous research findings [31], even though Finland as a country may have less diversity in parental education [61] than some other countries. However, this association should be critically evaluated because there was a bias towards highly educated parents in the present sample.

### 4.3. Gender Differences

In TGMD-3, boys had superior BS scores and a better gross motor index, while girls had better LM skills, as analyzed using a *t*-test. This finding is consistent with those of most previous studies comparing gender differences between girls and boys in BS [12,34,35,61] and LM skills [34,61,62], although a previous systematic review has stated no gender differences in LM skills [12] as well. Regarding TGMD-3, the findings of several studies are in concurrence with our findings that boys have a better gross motor index than girls [12,35]. In contrast, some studies have proclaimed that the gender differences may disappear on unifying LM skills and BS into a gross motor index [34,61]. Furthermore, our results indicate that because boys had higher BS scores than girls, boys may benefit from such unification as BS (54 points) can offer more points than LM skills (46 points) in the TGMD-3 gross motor index (100 points). As TGMD-3 is considered to be a sports-specific test, the hobbies and their content may reflect the development of MC. In fact, hobbies may differ between genders [19,27,35]. These aforementioned issues may explain the differences in MC measured using TGMD-3.

Previous studies have also shown that gender differences may become more evident if children do not participate in organized sports [58] or have lower MC [59]. Laukkanen et al. [59] found smaller gender differences in nationalities that have higher MC regardless of gender. Therefore, several studies have questioned whether the differences in MC may cease to exist in children aged under eight years if girls are provided with equivalent opportunities to practice sports [16,58]. Many researchers agree that gender differences during early childhood are not based on biological factors [15], but the differences seem more related to family, environmental, and sociocultural contexts [35,36,37,38], wherein girls tend to behave differently than boys [63,64].

When looking at the effect of gender in the regression models, there were no gender differences in TGMD-3, while there was a gender difference in KTK in Model 2. The result showed that being a girl was a positive predictor of a higher KTK total score. Previous studies [38,53] on balance and body coordination skills during early childhood revealed similar gender differences in some balance skills. However, the effect sizes were so small that gender could not be specified as an important correlate in children’s balance skills. Our research suggested that although gender (being a girl) was a positive predictor of the KTK total score in Model 2, the effect size was rather small; thus, greater gender differences occurred in TGMD-3 than in KTK. 

### 4.4. Strengths and Weaknesses of the Study

The strengths of this study included the use of two internationally well-known MC assessment tools offering more information about children’s MC [51,65], which is highly recommended [47,51]. The use of two test batteries made the results more robust and stronger. In addition, the study assessed a range of individual, family, and environmental correlates of MC based on the socioecological model. Moreover, the study sample was large, randomly selected, and nationally representative. However, the fact that MC assessment tools measure different aspects of MC can also be seen as a study limitation. Furthermore, the sample had an overrepresentation of highly educated parents, which is a limitation in many studies examining behaviors and attitude towards PA. The cross-sectional nature of our study also did not allow making any inference about causality. Furthermore, in the parental questionnaire, some test-retest reliability levels were slightly low with high 95% CIs. In the current study, we assessed the time spent outdoors and participation in organized sports using a proxy measure, which due to the age of the children could be considered a feasible method to assess the type and setting of PA. However, in future research, it would be optimal to combine parent-reported PA measures with device-based measures of PA such as accelerometers. Finally, in order to better understand the role of a child’s temperament in motor development and PA, observations of children’s motivation and persistence in physical play could be useful. Early educators could provide important information on children’s daily functioning.

## 5. Conclusions

The most important correlates associated with better MC were closely related to children’s biological (age, temperament traits such as activity and attention span-persistence) or behavioral (participation in organized sports) correlates. The TGMD-3 gross motor index explained 57% of the variance in MC, while the KTK total score explained 38% of the variance. Some small gender differences emerged in both test batteries, however, in a different manner. Therefore, it is important to note that the choice of test battery is crucial when investigating MC and gender differences. In sum, the findings may have relevant implications for both practical field study and research, especially in relation to the role of motor development. We state that skills are mainly learned by doing, and as a function of age, various movement experiences can be gained for example by participating in organized sports. In addition, the individual correlates such as temperament traits implied that to support young children’s MC development, and also to create efficient interventions to improve MC, it is useful to acknowledge the importance of individuality in learning. Thus, identifying individual (risk) factors, such as active, inattentive, or persistent temperament traits, may help in developing PA interventions that motivate also those children who might lag behind in age-appropriate motor skills. Moreover, the gender gap in motor learning needs to be explored in future research. On the other hand, early educators and parents should provide the same opportunities to be physically active and develop motor skills regardless of gender.

## Figures and Tables

**Figure 1 ijerph-17-02548-f001:**
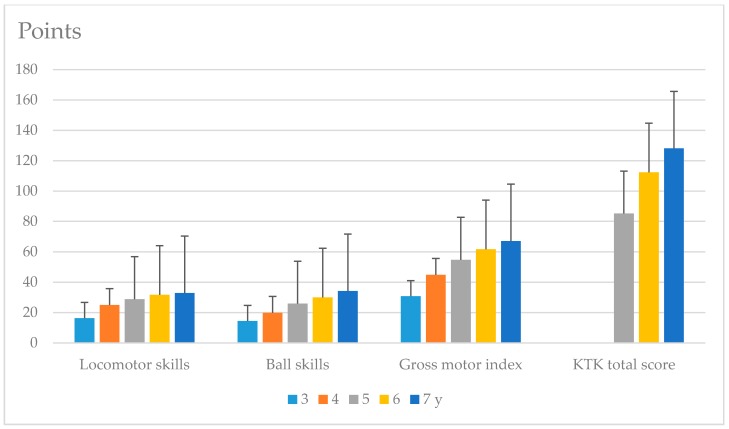
The level of motor competence (MC) measured using the Test of Gross Motor Development-Third Edition (TGMD-3) and Körperkoordinationstest für Kinder (KTK) assessment tools in the study sample (*n* = 945); values are reported as the mean and standard deviation (SD) scores for different ages (in y = years).

**Table 1 ijerph-17-02548-t001:** Descriptive statistics of the study sample (*n* = 945).

Age in Years (mean).						Total Sample		
Individual Correlates	3 (*n* = 116)	4(*n* = 227)	5(*n* = 244)	6(*n* = 295)	7(*n* = 63)	N	Mean (SD)	Gender Differences (*p-*Value)
Biological correlates								
Age (years)	3.54	4.48	5.48	6.46	7.13	945	5.42 (1.12)	0.36
- Girls	3.54	4.47	5.49	6.49	7.13	472	5.38 (1.13)	
- Boys	3.54	4.49	5.48	6.43	7.14	473	5.45 (1.11)	
Height (cm)	100.51	106.77	114.16	120.86	124.72	943	113.52 (9.73)	**0.01**
- Girls	100.47	106.41	113.14	120.35	124.05	472	112.64 (10.06)	
- Boys	100.55	107.23	115.10	121.30	125.46	471	114.40 (9.32)	
Weight (kg)	16.56	18.47	21.25	23.98	26.16	943	21.19 (4.47)	0.291
- Girls	16.36	18.39	21.35	23.91	26.32	472	21.04 (4.73)	
- Boys	16.76	18.57	21.15	24.04	25.97	471	21.34 (4.19)	
BMI SDS	0.26	0.19	0.18	0.16	0.26	943	0.19 (1.05)	0.61
- Girls	0.15	0.20	0.26	0.15	0.38	472	0.21 (1.13)	
- Boys	0.36	0.17	0.10	0.17	0.13	471	0.17 (0.98)	
Temperament								
Sociality (scale from 5 to 25 points)	17.22	18.17	18.53	18.38	19.20	929	18.28 (3.70)	0.82
- Girls	17.27	18.17	18.44	18.43	18.85	466	18.25 (3.63)	
- Boys	17.17	18.17	18.61	18.34	19.61	463	18.31 (3.77)	
Emotionality (scale from 5 to 25 points)	15.44	15.00	14.70	14.52	14.52	923	14.80 (3.11)	0.48
- Girls	15.45	14.70	14.85	14.36	14.6	461	14.72 (3.07)	
- Boys	15.43	15.40	14.55	14.67	14.39	462	14.87 (3.17)	
Activity (scale from 5 to 25 points)	18.90	19.09	18.48	18.66	18.08	914	18.71 (3.01)	**<0.001**
- Girls	18.75	18.54	17.63	18.11	17.48	460	18.14 (2.91)	
- Boys	19.05	19.82	19.30	19.12	18.79	454	19.28 (3.00)	
Attention span-persistence (scale from 5 to 25 points)	15.54	16.09	16.61	16.74	17.65	907	16.46 (3.01)	**0.01**
- Girls	15.53	16.43	16.85	17.21	17.48	455	16.73 (2.93)	
- Boys	15.54	15.65	16.37	16.35	17.85	452	16.20 (3.07)	
Reaction to food (scale from 5 to 25 points)	13.36	13.10	13.42	12.91	13.42	906	13.18 (4.44)	0.57
- Girls	13.00	13.17	13.37	13.20	13.94	452	13.26 (4.27)	
- Boys	13.70	13.01	13.48	12.68	12.76	454	13.09 (4.61)	
Soothability (scale from 5 to 25 points)	15.63	16.29	16.16	16.59	16.43	910	16.28 (3.26)	0.30
- Girls	15.47	16.48	16.12	16.85	16.75	457	16.39 (3.05)	
- Boys	15.79	16.04	16.21	16.37	16.07	453	16.17 (3.47)	
Behavioral correlates								
Sedentary time (min/day)	74.32	81.73	83.94	87.76	101.37	923	84.62 (47.31)	0.10
- Girls	72.17	76.69	81.52	89.72	88.13	459	82.01 (45.95)	
- Boys	76.32	88.03	86.16	86.03	115.50	464	87.20 (48.53)	
Time spent outdoors (scale from 1 to 7)	4.96	4.85	5.05	5.29	5.30	938	4.97 (1.19)	**0.001**
- Girls	5.00	4.80	4.87	5.16	5.24	469	5.22 (1.14)	
- Boys	4.97	4.97	5.22	5.45	5.37	469		
- Less than 1h/day (%)	13.2	8.0	13.2	8.6	6.3	94	10.0	
- Approximately 1 h/day (%)	51.8	66.4	51.0	43.8	50.8	493	52.6	
- 1-2 h/day (%)	35.1	25.7	35.8	47.6	42.9	351	37.4	
Participation in organized sports (min/week)	17.35	29.96	52.08	69.92	76.41	902	49.51 (65.28)	0.60
- Girls	23.02	31.66	56.54	64.55	59.58	448	48.34 (59.85)	
- Boys	12.26	27.86	47.94	74.64	95.00	454	50.65 (70.27)	
TGMD-3 locomotor skills (0 to 46 points)	16.33	24.97	28.84	31.66	32.87	945	27.52 (8.07)	**<0.001**
- Girls	17.65	26.44	30.53	33.39	33.24	472	28.89 (7.78)	
- Boys	15.05	23.07	27.29	30.15	32.47	473	26.16 (8.13)	
TGMD-3 ball skills (0 to 54 points)	14.47	19.89	25.84	29.99	34.24	945	24.87 (9.06)	**<0.001**
- Girls	13.12	18.43	23.14	27.40	30.94	472	22.43 (7.91)	
- Boys	15.76	21.77	28.32	32.23	37.87	473	27.29 (9.49)	
TGMD-3 gross motor index (0 to 100 points)	30.79	44.85	54.68	61.64	67.11	945	52.39 (15.16)	**0.030**
- Girls	30.77	44.87	53.67	60.80	64.18	472	51.32 (14.11)	
- Boys	30.81	44.84	55.61	62.38	70.33	473	53.46 (16.08)	
KTK (0 to 193 points)	-	-	85.20	112.25	128.14	416	103.21 (34.26)	0.19
- Girls	-	-	87.39	117.28	118.90	198	105.54 (33.51)	
- Boys	-	-	82.99	107.84	136.17	218	101.09 (34.86)	
**Family correlates**								
Parents’ education level (scale from 1 to 4)	2.78	2.73	2.79	2.66	2.75	935	100.0	0.99
- Girls	2.86	2.72	2.84	2.57	2.79			
- Boys	2.70	2.73	2.74	2.74	2.70			
- Elementary school (%)	4.4	4.4	1.7	2.1	1.6	26	2.8	
- Secondary/ vocational school (%)	28.1	29.6	30.7	36.8	33.3	301	32.2	
- Polytechnic (%)	35.1	38.5	36.9	38.1	38.1	351	37.5	
- University (%)	32.5	27.5	30.7	23.0	27.0	257	27.5	
Respondent’s physical activity (min/week)	59.19	56.76	58.06	58.42	57.20	845	57.93 (22.63)	0.88
- Girls	59.00	57.34	54.28	61.31	56.13	420	57.81 (21.12)	
- Boys	**59.38**	**56.01**	**61.55**	**55.99**	**58.42**	**425**	**58.06 (24.05)**	
**Environmental correlates**								
Electronic devices in use (*n*)	0.29	0.41	0.55	0.71	0.89	923	0.56 (0.92)	**0.04**
- Girls	0.31	0.35	0.49	0.63	0.78	463	0.49 (0.84)	
- Boys	0.26	0.47	0.60	0.79	1.00	460	0.62 (0.99)	
Access to sports facilities (scale from 0 to 44 points)	20.49	21.14	21.79	22.36	23.21	939	21.75 (4.17)	**0.04**
- Girls	20.91	21.01	21.37	21.74	23.39	467	21.47 (4.19)	
- Boys	20.10	21.31	22.18	22.90	23.00	472	22.03 (4.14)	
- Rarely (%)	27.1	21.4	16.4	10.4	5.5	141	16.4	
- Occasionally (%)	48.6	42.4	38.6	34.7	36.4	339	39.4	
- Weekly/daily (%)	24.3	36.2	45.0	54.9	58.2	380	44.2	

Statistically significant values are shown in bold. SD = standard deviation; BMI SDS = body mass index standard deviation scores; TGMD-3 = Test of Gross Motor Development-Third Edition; KTK = Körperkoordinationstest für Kinder.

**Table 2 ijerph-17-02548-t002:** Individual, family and environmental correlates associated with children’s TGMD-3 total score.

Variables	TGMD-3 Total Score				
	Model 1 (*n* = 716)			Model 2 (*n* = 856)	
Individual Correlates	Standardized *B* (95% CI)	*p*	* RE	Standardized *B* (95% CI)	*p*
Biological correlates					
- Age (months)	0.05 (0.60; 0.70)	<0.001		0.06 (0.61; 0.71)	<0.001
- Gender (1 = girls, 2 = boys)	0.02 (−0.02; 0.07)	0.32	6	0.03 (−0.01; 0.07)	0.19
- BMI SDS	−0.01 (−0.06; 0.04)	0.69	6		
Temperament (scale from 5 to 25 points in every subcategory mentioned below)					
- Sociability	0.04 (−0.02; 0.09)	0.20	7		
- Emotionality	0.01 (−0.05; 0.06)	0.81	4		
- Activity	0.11 (0.05; 0.16)	<0.01		0.11 (0.06; 0.15)	<0.001
- Attention span-persistence	0.04 (−0.01; 0.09)	0.12		0.05 (0.01; 0.10)	0.02
- Reaction to food	−0.04 (−0.09; 0.00)	0.07	9		
- Soothability	0 (−0.05; 0.06)	0.90	1		
Behavioral correlates					
- Sedentary time (min/day)	0.04 (−0.01; 0.09)	0.12	8		
- Time spent outdoors (scale from 1 to 7)	0.04 (−0.02; 0.10)	0.18	10		
- Participation in organized sports (min/week)	0.13 (0.07; 0.18)	<0.001		0.16 (0.12; 0.21)	<0.001
**Family correlates**					
- Parents’ education level (scale from 1 to 4)	0.01 (−0.04; 0.06)	0.72	3		
- Respondent’s physical activity (min/week)	0.01 (−0.04; 0.07)	0.65	5		
**Environmental correlates**					
- Electronic devices in use (n)	0 (−0.05; 0.05)	0.90	2		
- Access to sports facilities (scale from 0 to 44 points)	0.06 (−0.00; 0.11)	0.05		0.06 (0.01; 0.10)	0.03

Statistically significant values are shown in bold. 95% CI = confidence interval. * RE = removal order in which the explanatory variable was deleted from Model 1. In Model 2, only statistically significant correlates explaining the TGMD-3 gross motor index were left. In Models 1 and 2, the number of items varied because of missing data in the remaining variables. TGMD-3 = Test of Gross Motor Development-Third Edition; BMI SDS= body mass index standard deviation scores.

**Table 3 ijerph-17-02548-t003:** Individual, family and environmental correlates associated with children’s KTK total score.

Variables	KTK Total Score				
	Model 1 (*n* = 330)			Model 2 (*n* = 392)	
Individual Correlates	Standardized *B* (95% CI)	*p*	* RE	Standardized *B* (95% CI)	*p*
*Biological correlates*					
- Age (months)	0.51 (0.42; 0.59)	<0.001		0.50 (0.41; 0.58)	<0.001
- Gender (1 = girls, 2 = boys)	−0.08 (−0.16; 0.01)	0.07		−0.13 (−0.20; −0.05)	0.002
- BMI SDS	−0.01 (−0.12; 0.05)	0.42	8		
*Temperament* (scale from 5 to 25 points in every subcategory mentioned below)					
- Sociability	−0.08 (−0.17; 0.01)	0.09	9		
- Emotionality	−0.12 (−0.21; −0.03)	0.01		−0.12 (−0.20; −0.04)	0.003
- Activity	0.24 (0.16; 0.33)	<0.001		0.23 (0.15; 0.31)	<0.001
- Attention span-persistence	0.18 (0.10; 0.27)	<0.001		0.16 (0.08; 0.24)	<0.001
- Reaction to food	0.03 (−0.06; 0.11)	0.49	5		
- Soothability	0.04 (−0.06; 0.13)	0.43	7		
Behavioral correlates					
- Sedentary time (min/day)	0 (−0.09; 0.09)	0.96	1		
- Time spent outdoors (scale from 1 to 7)	0.01 (−0.08; 0.11)	0.81	3		
- Participation in organized sports (min/week)	0.13 (−0.03; 0.22)	0.01		0.12 (0.04; 0.21)	0.03
**Family correlates**					
- Parents’ education level (scale from 1 to 4)	0.08 (−0.01; 0.17)	0.08		0.09 (0.01; 0.17)	0.03
- Respondent’s physical activity (min/week)	0.01 (−0.08; 0.09)	0.88	2		
**Environmental correlates**					
- Electronic devices in use (n)	−0.06 (−0.15; 0.02)	0.15	4		
- Access to sports facilities (scale from 0 to 44 points)	0.04 (−0.07; 0.14)	0.49	6		

Statistically significant values are shown in bold. *RE = Removal order in which the explanatory variable was deleted from model 1. In model 2, only statistically significant factors explaining the KTK result were left. In models 1 and 2, the number of items varied because of missing data in the remaining variables. KTK = Körperkoordinationstest für Kinder; BMI SDS = body mass index standard deviation scores.

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
