# Peer review of "Individual, Family, and Environmental Correlates of Motor Competence in Young Children: Regression Model Analysis of Data Obtained from Two Motor Tests"

_ijerph, 2020, doi:10.3390/ijerph17072548_

Round 1

Reviewer 1 Report

General

This is a very well written study which presents some interesting and useful information and I commend the authors on producing this work. Results are well presented and the analysis is appropriate. I have some minor comments below that the authors should consider when making any revision the editor permits them to.

Specific

Abstract first line: Please modify this, the strength of the statement is overly strong for the extant evidence on this topic.

Introduction Line 34-35: the authors need to be clear here that the relationship between PA and MC is bidirectional. As this is written it suggests that PA leads to lower MC. I don’t disagree in early childhood but for older children the Stodden model suggests this is not necessarily the case.

Line 67-68: I think the authors need to add something after the word ‘category’. An eg perhaps? To better clarify the statement.

Lines 91-95: Could the authors here re-emphasise the gap in the literature their study will address?

Line 150: ‘made to complete’ sounds a little strange, I assume the agreed to complete it and participated, so perhaps change the way this is written

Discussion, Lines 294-319: it would be useful here to add an overarching statement identifying the practical application of their overall findings here.

In the limitations section, it would also be useful to add mention that physical activity was not actually assessed, rather self reported aspects related to physical activity were assessed and it was important to do so, however, future work using object assessments of PA might be useful?

Author Response

This is a very well written study which presents some interesting and useful information and I commend the authors on producing this work. Results are well presented and the analysis is appropriate. I have some minor comments below that the authors should consider when making any revision the editor permits them to.

Specific

We value the time that you, the editors and reviewers, have taken to review our manuscript, and we thank you for your insightful comments, which have helped us improve our paper. Please find below our point-by-point responses to your comments written in red. Related modifications to the manuscript have been highlighted in yellow.

Before we proceed to more detailed comments, we would like to point out that we have made some minor changes to the manuscript in response to your relevant comments. First, the reference number 66 was deleted (line 608) as it was not cited in the text. We paid extra attention to abstract (lines 13-29), Strengths and weaknesses of the study (lines 414-430) and conclusions (lines 432-448). 

ABSTRACT

Abstract first line: Please modify this, the strength of the statement is overly strong for the extant evidence on this topic.

We agree that the phrase was strongly stated. We have modified this as follows (lines 13-14): “Physical activity and motor competence (MC) have been considered to be closely related and preventing childhood obesity.”

Introduction Line 34-35: the authors need to be clear here that the relationship between PA and MC is bidirectional. As this is written it suggests that PA leads to lower MC. I don’t disagree in early childhood but for older children the Stodden model suggests this is not necessarily the case.

We have modified this to make the bidirectional association between MC and PA clearer (in lines 34-38):

“Motor competence (MC) and physical activity (PA) have been found to be closely and  bidirectionally related in several theoretical models [1–3] and studies [4]. Consequently, it is claimed that if a child is not physically active, they may have a greater tendency to have lower MC, or vice versa, and be at risk of gaining unhealthy body weight [5]. Children’s weight status negatively influences their future level of gross motor coordination, and vice versa [6].”

Line 67-68: I think the authors need to add something after the word ‘category’. An eg perhaps? To better clarify the statement.

You can find the modified phrase in lines 69-70: “As motor development is a multidimensional process [11,27], it differs according to the motor skill category e.g. [27,28].”

Lines 91-95: Could the authors here re-emphasise the gap in the literature their study will address?

We have added in the end of the introduction section a sentence which targets to emphasize the gap between the previous studies and on the other hand, the novel information that the current study aims to add into research (lines 96-99) “As majority of the studies [12, 34, 35, 36] focus on individual correlates of MC, we wanted to broaden the research into socioecological perspective, such as family and environmental correlates to gain more knowledge about MC development.”

Line 150: ‘made to complete’ sounds a little strange, I assume the agreed to complete it and participated, so perhaps change the way this is written

We modified the phrase as follows (lines 153-154): “To have complementary information about gross motor coordination and body control of children aged 5 to 7 years, they participated also in the KTK assessment.”

Discussion, Lines 294-319: it would be useful here to add an overarching statement identifying the practical application of their overall findings here.

We agree that practical application should be added in the text. However, we decided to locate these in the conclusion part (in lines 438-448) as it seems to be common manner in the published articles in the current journal:

“In sum, the findings may have relevant implications for both the practical field and research, especially in relation to the role of motor development. We state that skills are mainly learnt by doing, and as a function of age various movement experiences can be gained for example by participating in organised sport. In addition, the individual correlates such as temperament traits imply that to support young children’s MC development, and also to create efficient interventions to improve MC, it is useful to acknowledge the importance of individuality in learning. Thus, identifying individual (risk) factors, such as active, inattentive or persistent temperament traits, may help in developing PA interventions that motivate also those children who might lack behind in age appropriate motor skills. Moreover, the gender gap in motor learning needs to be explored in future research. On the other hand, early educators and parents should provide the same opportunities to be physically active and develop motor skills regardless of the gender.”

In the limitations section, it would also be useful to add mention that physical activity was not actually assessed, rather self reported aspects related to physical activity were assessed and it was important to do so, however, future work using object assessments of PA might be useful?

This limitation and suggestion for future research has been added in the text (4.4. Strengths and weaknesses of the study section, lines 424-427):

“In the current study, we assessed the time spent outdoors and participation in organised sports using a proxy measure, which due to the age of the children can be considered a feasible method to assess the type and setting of PA. However, in future research it would be optimal to combine parent-reported PA measures with device-based measures of PA such as accelerometers.”

Reviewer 2 Report

The article titled "Individual, Family and Environmental Correlates of Motor Competence in Young Children: Regression Model Analysis of Data Obtained from Two Motor Tests" describes very important research in the context of planning healthy development of children aged 3-7 years. In my opinion, the study was well planned, well carried out and well described. The noted differences between girls and boys are minimal, and their interpretation is correct. The presented conclusions indicate the direction of further research. In the next study, if it is planned, it is worth choosing a research sample so that it takes into account parents with different levels of education in equal proportions or in proportions representative of the population.

Author Response

The article titled "Individual, Family and Environmental Correlates of Motor Competence in Young Children: Regression Model Analysis of Data Obtained from Two Motor Tests" describes very important research in the context of planning healthy development of children aged 3-7 years. In my opinion, the study was well planned, well carried out and well described. The noted differences between girls and boys are minimal, and their interpretation is correct. The presented conclusions indicate the direction of further research. In the next study, if it is planned, it is worth choosing a research sample so that it takes into account parents with different levels of education in equal proportions or in proportions representative of the population.

We value the time that you, the editors and reviewers, have taken to review our manuscript, and we thank you for your support for this manuscript. Some minor changes have been done in the manuscript and you can find them in the text highlighted in yellow.

As a summary, we have made some minor changes to the manuscript in response to all reviewers’ relevant comments. First, the reference number 66 was deleted (line 608) as it was not cited in the text. We paid extra attention to abstract (lines 13-29), Strengths and weaknesses of the study (lines 414-430) and conclusions (lines 432-448). 

Reviewer 3 Report

Manuscript ID: ijerph-765093

The manuscript is relevant and advances knowledge to merit publication, but I would like to suggest some minor issues:

ABSTRACT

The aims of the study should be shown in the abstract.

MATERIALS AND METHODS

I think Table 1 should be in the results section.

DISCUSSION

The authors should recognize as a weakness of their study the possibility that the assessment tools measure different aspects of motor competence. This can be the main limitation.

In addition, it would be useful to indicate future research and practical applications of the study.

CONCLUSIONS

The conclusions of the study must be related to the objectives of the research. In this sense, is important to explain something else about gender differences. The same in the abstract.

REFERENCES

The reference 66 is missing.

Author Response

The manuscript is relevant and advances knowledge to merit publication, but I would like to suggest some minor issues:

We value the time that you, the editors and reviewers, have taken to review our manuscript, and we thank you for your insightful comments, which have helped us improve our paper. Please find below our point-by-point responses to your comments written in red. Related modifications to the manuscript have been highlighted in yellow.

Before we proceed to more detailed comments, we would like to point out that we have made some minor changes to the manuscript in response to your relevant comments. First, the reference number 66 was deleted (line 608) as it was not cited in the text. We paid extra attention to abstract (lines 13-29), Strengths and weaknesses of the study (lines 414-430) and conclusions (lines 432-448). 

ABSTRACT

The aims of the study should be shown in the abstract.

The aims of the study have been added in the text (lines 14-16): “The aim of the study was two-fold: to examine MC measured with two different tools in relation to individual, family and environmental correlates, and to investigate gender differences in MC.”

MATERIALS AND METHODS

I think Table 1 should be in the results section.

Thank you for your opinion. We agree that the Table 1 could have been replaced in the results section as well. However, we did this choice by putting the Table 1 already in the methods section as it was what the journal guidelines asked us to do when preparing the manuscript. The instructions say:

“All Figures, Schemes and Tables should be inserted into the main text close to their first citation and must be numbered following their number of appearance (Figure 1, Scheme I, Figure 2, Scheme II, Table 1, etc.).”

Source: https://www.mdpi.com/journal/ijerph/instructions#figures

Therefore, as we cite the Table 1 already in the methods section, we believe that it is better to keep it where it was originally due to the guidelines of the journal.

DISCUSSION

The authors should recognize as a weakness of their study the possibility that the assessment tools measure different aspects of motor competence. This can be the main limitation.

We agree that the two internationally well-known MC assessment tools are used, it can be the main strength or the main limitation of the study. Therefore we have added in the 4.4. Strengths and weaknesses of the study –paragraph (in lines 419-420) a phrase saying: “However, the fact that MC assessment tools measure different aspects of MC can also be seen as a study limitation.”

In addition, it would be useful to indicate future research and practical applications of the study.

We have aimed to better emphasize the future research, see following sentences in the discussion:

  • Lines 354-356: “The future research would benefit from multidisciplinary collaboration between different professionals to better understand the role of a child’s temperament and parents’ behaviour in the development of MC.”
  • Lines 424-427: “In the current study, we assessed the time spent outdoors and participation in organised sports using a proxy measure, which due to the age of the children can be considered a feasible method to assess the type and setting of PA. However, in future research it would be optimal to combine parent-reported PA measures with device-based measures of PA such as accelerometers.”
  • Lines 428-430: “Finally, in order to better understand the role of a child’s temperament in motor development and PA, observations on children’s motivation and persistence in physical play could be useful. Early educators could provide important information on children’s daily functioning.”

Practical applications of the study are also presented in the conclusions:

  • Lines 437-448: “In sum, the findings may have relevant implications for both the practical field and research, especially in relation to the role of motor development. We state that skills are mainly learnt by doing, and as a function of age various movement experiences can be gained for example by participating in organised sport. In addition, the individual correlates such as temperament traits imply that to support young children’s MC development, and also to create efficient interventions to improve MC, it is useful to acknowledge the importance of individuality in learning. Thus, identifying individual (risk) factors, such as active, inattentive or persistent temperament traits, may help in developing PA interventions that motivate also those children who might lack behind in age appropriate motor skills. Moreover, the gender gap in motor learning needs to be explored in future research. On the other hand, early educators and parents should provide the same opportunities to be physically active and develop motor skills regardless of the gender.”

CONCLUSIONS

The conclusions of the study must be related to the objectives of the research. In this sense, is important to explain something else about gender differences. The same in the abstract.

As the abstract word limitation is 200, we believe that all cannot be said. However, as the gender differences was the second aim of the study, it should be mentioned in the abstract, we agree. Thank you for noticing this. Indeed, we added gender differences so that it is mentioned as an aim of the study, and second, also the results the differences are written already in the abstract. Moreover, we paid attention to results of the gender differences in conclusion of the study.

Abstract

  • Lines 26-27: “Small gender differences were found in both assessment tools, albeit in different manner.”

Related to conclusions, we added some information about the gender differences. Nevertheless, we did not want to emphasize this result too much, as the results of the gender differences were different and small. We believe that based on the results, to emphasize too much or to draw strong conclusions would be inappropriate.

Conclusions

  • Lines 435-437: “Some small gender differences emerged in both the test batteries, however, in a different manner. Therefore, it is important to note that the choice of test battery is crucial when investigating MC and gender differences.”

Gender differences are also taking in to account in the practical implications of the study. See below.

Practical implications

  • Lines 445-448: “Moreover, the gender gap in motor learning needs to be explored in future research. On the other hand, early educators and parents should provide the same opportunities to be physically active and develop motor skills regardless of the gender.”

 REFERENCES

The reference 66 is missing.

Thank you for your careful work. We have removed the reference 66 from the references as it was not mentioned in the text. See line 608.